# Deciphering the Molecular Mechanisms of Autonomic Nervous System Neuron Induction through Integrative Bioinformatics Analysis

**DOI:** 10.3390/ijms24109053

**Published:** 2023-05-21

**Authors:** Yuzo Takayama, Yuka Akagi, Yasuyuki S. Kida

**Affiliations:** 1Cellular and Molecular Biotechnology Institute, National Institute of Advanced Industrial Science and Technology (AIST), Central 5-41, 1-1-1 Higashi, Tsukuba 305-8565, Japan; y-akagi@aist.go.jp; 2Tsukuba Life Science Innovation Program (T-LSI), School of Comprehensive Human Sciences, University of Tsukuba, 1-1-1 Tennoudai, Tsukuba 305-8572, Japan; 3School of Integrative & Global Majors, University of Tsukuba, Tsukuba 305-8572, Japan

**Keywords:** autonomic nervous system, bioinformatics, human induced pluripotent stem cells

## Abstract

In vitro derivation of human neurons in the autonomic nervous system (ANS) is an important technology, given its regulatory roles in maintaining homeostasis in the human body. Although several induction protocols for autonomic lineages have been reported, the regulatory machinery remains largely undefined, primarily due to the absence of a comprehensive understanding of the molecular mechanism regulating human autonomic induction in vitro. In this study, our objective was to pinpoint key regulatory components using integrated bioinformatics analysis. A protein–protein interaction network construction for the proteins encoded by the differentially expressed genes from our RNA sequencing data, and conducting subsequent module analysis, we identified distinct gene clusters and hub genes involved in the induction of autonomic lineages. Moreover, we analyzed the impact of transcription factor (TF) activity on target gene expression, revealing enhanced autonomic TF activity that could lead to the induction of autonomic lineages. The accuracy of this bioinformatics analysis was corroborated by employing calcium imaging to observe specific responses to certain ANS agonists. This investigation offers novel insights into the regulatory machinery in the generation of neurons in the ANS, which would be valuable for further understanding and precise regulation of autonomic induction and differentiation.

## 1. Introduction

The autonomic nervous system (ANS) governs homeostasis and unconscious body functions through the opposing regulations of sympathetic and parasympathetic neurons, which are the two major components of the ANS. Therefore, an imbalance between these neuronal activity leads to various dysfunctional symptoms, such as hypertension, irregular heart rate, and respiratory failure. The dysfunctional symptoms have been found in a variety of diseases: familial dysautonomia [1], diabetic neuropathy [2], Parkinson’s disease [3], Alzheimer’s disease [4], and COVID-19 [5]. ANS activity also has been shown to be involved in the regulation of tumor microenvironments [6] and organ development [7]. Consequently, there are critical needs for technologies such as generation of human sympathetic and parasympathetic neurons as well as reconstruction of the ANS in vitro from the perspective of both basic medical research and translational research, such as drug discovery and regenerative medicine.

Human induced pluripotent stem cell (iPSC) technology [8,9] provides promising experimental platforms to mimic in vivo human physiology and study the dysfunction of human organs using disease-specific iPSCs. Several groups, including ours, have reported induction protocols for producing ANS progenitors [10,11,12,13]. The evaluation of induced ANS progenitors often involves analysis of the expression of key transcription factor (TF) ASCL1 and PHOX2B [14,15]. In particular, PHOX2B is a crucial marker for ANS progenitors, known to regulate the expression of TH and DBH, which are sympathetic neuron markers, and CHAT, a parasympathetic neuron marker [16]. These induction protocols were based on different approaches such as controlling signaling modulations or cell isolation techniques. Oh et al. have established a derivation protocol of sympathetic progenitors from human pluripotent stem cells (PSCs) through WNT signaling activation, followed by activation of sonic hedgehog (SHH) and bone morphogenetic protein (BMP) signaling in combination with NOTCH and vascular endothelial growth factor receptor inhibitors [10]. Frith et al. also have established a different method of a sympathoadrenal progenitor derivation from human PSCs through the combination of WNT signaling activation and moderate BMP activation, followed by SHH signaling activation [11]. Additionally, Kirino et al., have established a derivation protocol of sympathetic neuronal progenitors from human PSCs through WNT signaling activation, but followed by BMP activation and application of retinoic acid, which mediates anterior/posterior patterning [12]. We have also derived autonomic lineage progenitors from human PSCs through WNT signaling activation followed by inhibition of WNT and SHH signaling, and BMP signaling activation [13]. These reports have raised the possibility that the diversity of autonomic lineages in vitro is due to different method-induced progenitors and their derivative neurons. Alternatively, it could be that a definitive induction protocol, which accurately replicates the differentiation processes of in vivo autonomic lineages, has not yet been provided.

Recent rapid advancements in RNA-sequencing (RNA-seq) technologies have enabled fast and genome-wide profiling not only for mRNA, but also for microRNA (miRNA) [17], long non-coding RNA (lncRNA) [18], and TF biding regions [19]. Bioinformatic data mining and gene regulatory network (GRN)-based analysis offer identification and prediction of important key genes and pathways involved in the developmental processes of cells or tissues [20,21]. In this study, time-pointed RNA-seq data obtained from differentiating stage of cells toward the progenitors in the ANS were analyzed to identify differentially expressed genes (DEGs). The DEGs were then used to construct protein–protein interaction (PPI) networks, which were analyzed using a module analysis approach. In addition, another PPI network was constructed to identify genes associated with the development of sympathetic and parasympathetic neurons within the ANS progenitors, and to explore the regulatory components involved in this process. The analysis also suggested the possibility that unidentified regulatory mechanisms, such as those involving TF activity, could play a role in the induction of these neurons.

## 2. Results

### 2.1. Stepwise Chemical Treatment Leads to Autonomic Development from Human PSCs

An RNA-seq data analysis focused on developing an induction method for generating ANS progenitors from human PSCs [13] was performed to obtain a comprehensive understanding of the differentiation processes involved.

Autonomic induction of human PSCs was conducted by sequentially exposing them to dorsomorphin (DM; a BMP signaling inhibitor), SB431542 (SB; a TGF-β signaling inhibitor), CHIR99021 (CHIR; a WNT signaling activator), IWR1 (a WNT signaling inhibitor), SANT1 (a SHH signaling inhibitor), and BMP4 over a period of 13 days under floating culture conditions (Figure 1A). Using this method, compared to PSCs, the induced cells strongly expressed neural crest (NC) marker SOX10, peripheral neuron marker PRPH, autonomic lineage markers PHOX2A and PHOX2B, and sympathetic neuronal marker TH (Figure 1B) on day 13.

Subsequently, the induced progenitors were further differentiated and matured into functional neurons. These neurons were spontaneously formed network structures with ganglion-like organization (Figure 1C). The generation of sympathetic neurons marked with DBH/PRPH and parasympathetic neurons marked with PHOX2B/CHAT was also confirmed (Figure 1D,E). To confirm the functionalities of the neurons, cells were plated onto microelectrode arrays (MEAs) to analyze their electrical activity. The frequency of spike activity was enhanced by the application of nicotine, which activates the nicotinic acetylcholine receptors (nAChRs) of post-ganglionic sympathetic or parasympathetic neurons. (Figure 1F). These results verified that our induction protocol successfully directed autonomic induction of human PSCs and generated sympathetic and parasympathetic neurons.

### 2.2. Global Transcriptome Changes Were Revealed during Autonomic Induction

First, the RNA-seq data from day 0 and day 13 samples were compared to summarize the global transcriptome changes occurring during autonomic induction (Figure 2A). A total of 3926 upregulated and 3878 downregulated DEGs were identified in the day 13 samples compared to the day 0 sample. To estimate the functional involvement of these DEGs, the top three representative enriched pathways from the gene ontology biological process (GO_BP), Kyoto Encyclopedia of Genes and Genomes (KEGG) pathways [22], Reactome pathways [23], and Wikipathways [24] were depicted. The upregulated DEGs were primarily enriched in neuronal differentiation- or neuronal development-associated pathways through the GO_BP, KEGG pathway, and Reactome pathways, as well as in NC differentiation, according to Wikipathways (Figure 2B). In contrast, the downregulated DEGs were enriched in diverse pathways, such as the apoptosis, cell cycle, and metabolic pathways, reflecting the exit from an undifferentiated state (Figure 2C). Additionally, all the enriched terms of GO_BP and the pathways are listed in the Appendix A. A gene set enrichment analysis (GSEA) was also performed for the upregulated DEGs, and it was revealed that “autonomic nervous system development (GO_BP)” was enriched in day 13 samples. Notably, important regulators for autonomic development, such as ASCL1, GATA3, PHOX2A, and PHOX2B [10,11,12,13], were included in the GSEA outcome (Figure 2D). These global transcriptome results further supported that our protocol successfully initiated autonomic induction.

### 2.3. Gene Regulatory Network Analysis Identifies Key Gene Clusters and Hub Genes

To further investigate the effects of stepwise induction, RNA-seq data were analyzed for upregulated DEGs at an early stage (day 0 and day 7) and a later stage (day 7 and day 13). A total of 3063 upregulated DEGs were identified in the day 7 samples compared to day 0 (Figure 3A). Subsequently, a GRN analysis based on PPI construction, molecular complex detection (MCODE) module analysis, and hub gene detection were performed, identifying four key gene clusters involved in our induction processes. The MCODE analysis identified 17 gene clusters from the PPI networks. Among them, the top four gene clusters had the highest MCODE scores, indicating most interconnected network component, were chosen for further analysis. Detected genes in cluster 1 and 3 were primarily enriched in cell organelle-associated pathways (Figure 3B,D). In cluster 2, 97 detected genes, including WNT genes (WNT1, WNT3A, and WNT5A) as hub genes, were markedly enriched in cell differentiation-associated pathways such as “Neural crest differentiation (Wikipathways)” (Figure 3C). In cluster 4, 144 detected genes, including 5 hub genes (CDKN1C, IKBKB, MAP2, PPP2R4, and XIAP), were markedly enriched in nervous system-associated pathways (Figure 3D). Additionally, all enriched terms of GO_BP and pathways in each gene cluster are listed in the Appendix A. These findings suggest that the stepwise signaling modification protocol induced human PSCs into NC lineage cells during the first step of induction, as indicated by upregulation of the key gene clusters and signaling pathways associated with autonomic induction [13].

In the analysis of the later step, a total of 2304 upregulated DEGs were identified on day 13 compared to day 7 (Figure 4A). Subsequently, the MCODE analysis identified 10 gene clusters from the PPI networks, with the top four gene clusters having the highest MCODE scores. The GRN analysis revealed that key gene clusters were primarily associated with neuronal structure- and nervous system-related pathways. In cluster 1, 53 detected genes, including 5 hub genes (GRIA2, NRXN1, SYT1, SYT4, and VAMP2), were markedly enriched in synapse-related pathways (Figure 4B). In cluster 2, 168 detected genes, including 5 hub genes (CAMK2A, MAPT, SLC17A6, SLC17A7, and SYP), were markedly enriched in axon-related pathways (Figure 4C). In cluster 3, 211 detected genes, including 5 hub genes (CASP3, HRAS, KDR, PIK3R1, and SOX9), were markedly enriched in nervous differentiation-associated pathways (Figure 4D). In cluster 4, 55 detected genes, including 5 hub genes (HIST1H2AC, HIST1H2BD, HIST1H3J, HIST2H3A, and SNAP25), were enriched in BMP signaling-associated pathways (Figure 4E). Additionally, all the enriched terms of GO_BP and pathways in each gene cluster are listed in Appendix A. These findings suggest that the stepwise signaling modification protocol promoted neuronal differentiation of induced NC lineage cells during the latter step of the induction.

### 2.4. Core Gene Network Analysis and Functional Evaluation of Autonomic Induction

To identify further regulatory components for autonomic induction, a PPI network was constructed using the seven important regulators for autonomic development, ASCL1, DDC, GATA2, GATA3, PHOX2A, PHOX2B, and TH, which were previously found to be upregulated DEGs on day 13 compared to day 0 [13]. By constructing the PPI network, a total of 101 genes were identified, including these regulators. These genes were linked by two or more associations with each other in the PPI network (Figure 5A). In addition to these input genes, significant autonomic key-functional genes such as CTNNB1, DBH, HPCAL1, ID2, INSM1, ISL1, MEIS1, NF1, and SLC18A3 were newly identified as components of our induction protocol. Surprisingly, sensory nervous system lineage TF genes, such as POU4F1, NEUROD1, and NEUROG1, were also detected. To investigate in more detail, the expression patterns of the TF-target genes were analyzed as an indicator of TF regulatory activity [25] (Figure 5B). As representative TFs, ASCL1 and PHOX2B were selected for autonomic TFs, and NEUROD1 and POU4F1 were selected for sensory TFs [25]. Then, expression pattern changes of TF-target genes, with predicted TF binding motifs, were analyzed and visualized from day 0 to day 13. Genes targeted by autonomic TFs exhibited an upregulation trend, while genes targeted by sensory TFs displayed a random expression pattern. These lines of evidence indicate that the regulatory network induced by our induction protocol highly enhances autonomic TF activity and not sensory TF activity. The accuracy of this bioinformatics analysis regarding TF activation can be confirmed by using calcium imaging to observe the specific response to certain agonists, which is characteristic of the autonomic nervous system. Evoked responses in calcium signals were analyzed by applying L-glutamate, a general excitatory neurotransmitter, or nicotine, which activates nAChRs in post-ganglionic sympathetic or parasympathetic neurons, on day 33. The induced neurons exhibited strong responses to both L-glutamate and nicotine, confirming that they were predominantly derived from the autonomic lineage (Figure 5C, Appendix A). In contrast, these neurons did not show distinct responses to sensory neuron agonists such as capsaicin or menthol (Figure 5D, Appendix A). These results support the integrative data analysis gene, despite the upregulation of sensory marker genes.

## 3. Discussion

In this study, we conducted an integrative bioinformatics analysis to decipher the molecular mechanisms of autonomic nervous system neuron induction from human PSCs. Global transcriptome analysis revealed the upregulation of neuronal differentiation and development-associated pathways during autonomic induction. Notably, induced progenitors exhibited upregulated expressions of important TF genes in ANS development, such as *ASCL1* and *PHOX2B*, via NC lineage stages. GRN analysis further identified major gene clusters, hub genes, and enriched pathways associated with cell differentiation, nervous system development, and neuronal structure during the early and later steps of autonomic induction. Induced autonomic neurons have a myriad potential applications in biomedical research. The most immediately accessible application is due to their ability to facilitate disease modeling, contributing significantly to our understanding of disease pathophysiology. This, in turn, propels advancements in drug discovery and validation. Furthermore, these neurons can be leveraged in the fields of regenerative medicine and gene therapy research, acting as valuable therapeutic tools or resources. Collectively, these applications have substantial implications for the evolution of personalized medicine, shaping a future wherein treatments are tailored to individual patient profiles. For use as research tools in modeling, the neurons would allow us to construct co-culture systems with various target organs. This could potentially reconstitute the homeostatic functions and inter-organ interactions in vitro. This approach could further contribute to unraveling the mechanisms of diseases involving the ANS, and could be applied to drug screening.

Module analysis and subsequent hub gene identification of the PPI network of upregulated DEGs can identify key gene clusters. At the early stage (day 0 to day 7), the identified gene clusters and hub genes were associated with NC and nervous system differentiation pathways (Figure 3), suggesting that cells on day 7 were already differentiated into NC cells. The hub genes in cluster 2 include distinct WNT genes, which also support the induction of NC cells [26,27]. Conversely, in the subsequent phase (day 7 to day 13), these genes were significantly associated with nervous system-related pathways (Figure 4), implying that the induced NC cells underwent differentiation into neurons during this timeframe. In addition, the enriched GO terms included ANS-related terms, such as the “norepinephrine neurotransmitter release cycle (Reactome Pathways)” and the “BMP signaling pathway (GO Biological Processes)”, indicating that induced cells acquired the characteristics of autonomic lineages. The identification of hub genes could potentially provide beneficial biomarkers for the induction of the ANS in vitro. To validate this possibility, it would be intriguing to conduct further research involving experiments on the identified hub genes or other notable genes within the gene clusters [28,29].

The constructed PPI network indicated core regulatory components in our induction protocol. The seven input genes were associated with other autonomic marker genes listed in the results section, supporting the intended differentiation toward autonomic lineages. Two distinct types of NC-markers were also identified: *ETS1*, *MSX1*, *MSX2* and TFAP2B for cranial NC cells, and *DLL1* and *MOXD1* for trunk NC cells. Given that cranial and trunk NC cells and their derivatives are the sources of parasympathetic and sympathetic neurons during development [30,31,32], respectively, these gene expression patterns indicate that our induction protocols follow differentiation processes in vivo. Intriguingly, our comprehensive RNA-seq analysis unveiled elevated expressions of sensory marker genes. However, the TF activity analysis demonstrated that autonomic TFs tended to enhance the expressions of their target genes, potentially leading to the selective induction of sympathetic and parasympathetic neurons. This finding suggests that the autonomic induction may be regulated by intricate mechanisms involving miRNAs [33,34] and lncRNAs [35,36], which can modulate TF functions. Although miRNAs and lncRNAs have been implicated in central nervous system development and neurodegenerative diseases [37,38], their roles in ANS development remain largely unexplored, with a few exceptions highlighting their regulatory roles in peripheral nervous system injury and regeneration [39,40]. Further investigations into the roles of miRNAs and lncRNAs during autonomic induction will help unravel the mechanisms underlying ANS development.

On the other hand, a limitation and challenge of this study is the difficulty associated with obtaining human sympathetic and parasympathetic neurons for transcriptome analysis. This difficulty is presumed to arise from the fact that sympathetic and parasympathetic neurons are dispersed throughout the body, are present in limited numbers, and are also subject to ethical considerations. Nonetheless, it is important to compare and analyze the transcriptome of actual human sympathetic and parasympathetic neurons, and induced neurons from human PSCs. Thus, we believe it is crucial to explore the feasibility of collecting and analyzing autonomic lineages in future studies.

In conclusion, we provide a comprehensive understanding of the molecular mechanisms underlying ANS neuron induction from human PSCs, and the identified core gene networks and regulatory components could be valuable for the development of fine-tuning induction protocols as well as novel therapeutic strategies targeting autonomic disorders. Further validation and exploration of the underlying machinery involved in autonomic induction will contribute to a deeper understanding of human ANS development and ANS-related disorders.

## 4. Materials and Methods

### 4.1. Cell Culture

Human iPSC line 201B7 (female) was obtained from RIKEN BioResource Research Center (RIKEN BRC; Tsukuba, Japan). Human embryonic stem cell (ESC) line H1 (male) was obtained from Wicell Research Institute (Madison, WI, USA). Maintenance of the human PSCs and autonomic induction were performed according to previously described protocols [13]. Briefly, the human PSCs were harvested and cultured in mTeSR1 WO 2ME/MV medium (STEMCELL Technologies) on Laminin511-E8 (iMatrix511; Nippi, Tokyo, Japan)-coated plates. The culture medium was changed daily. Cells were passaged with Accutase (Thermo Fisher Scientific, Waltham, MA, USA) and ROCK inhibitor Y-27632 (10 µM, FUJIFILM Wako Pure Chemical Industries, Osaka, Japan) was added to the culture medium for the first 24h after plating.

For autonomic induction, collected cells were transferred into 6-well plates (Corning Inc., Corning, NY, USA) coated with 2-methacryloyloxyethyl phosphorylcholine (MPC) polymer (Lipidure CM5206E; NOF, Tokyo, Japan) at a density of 1–2 × 10^5^ cells/cm^2^. The MPC polymer-coated plates were incubated at 95 rpm on a rotary shaker (OS-762RC, Optime, Tokyo, Japan). Cells were maintained in mTeSR1 WO 2ME/MV medium containing 10 µM Y-27632 for 3 days to form EBs. To initiate induction, the EBs were cultured in knockout serum replacement (KSR) medium (DMEM-F12 (FUJIFILM Wako Pure Chemical Industries), 20% KSR (Life Technologies, Carlsbad, CA, USA), 1% non-essential amino acids (FUJIFILM Wako Pure Chemical Industries), 1% monothioglycerol (FUJIFILM Wako Pure Chemical Industries), and 1% penicillin-streptomycin (FUJIFILM Wako Pure Chemical Industries)) containing dorsomorphin (2 µM; DM; Sigma-Aldrich, St. Louis, MO, USA), SB431542 (10 µM; SB; Sigma-Aldrich), and bFGF (10 ng/mL; FUJIFILM Wako Pure Chemical Industries) for 2 days (day 0–2). On day 2–5, the EBs were cultured in KSR medium containing CHIR99021 (3 µM; CHIR; Cayman Chemical, Ann Arbor, MI, USA), SB (20 µM), and bFGF (10 ng/mL). On day 5–7, the EBs were cultured in a 3:1 mixture of KSR and N2 medium (DMEM-F12, 1% N2 supplement (FUJIFILM Wako Pure Chemical Industries), 1% non-essential amino acids, and 1% penicillin-streptomycin containing CHIR (3 µM) and bFGF (10 ng/mL). On day 7–9, the EBs were cultured in a 1:1 mixture of KSR/N2 medium containing IWR1 (10 µM; Sigma-Aldrich), SANT1 (250 nM; Sigma-Aldrich), recombinant human BMP4 (25 ng/mL; FUJIFILM Wako Pure Chemical Industries), and bFGF (10 ng/mL). From day 9–13 (changing the medium on day 12), the EBs were cultured in a 1:3 mixture of KSR/N2 medium containing IWR1 (10 µM), SANT1 (250 nM), BMP4 (25 ng/mL), and bFGF (10 ng/mL).

Human PSC experiments were approved by the Ethics Committee of the National Institute of Advanced Industrial Science and Technology (AIST). The use of human ESCs was approved by the Ministry of Education, Culture, Sports, Science, and Technology (MEXT) of Japan. Furthermore, the use of human ESCs was carried out in accordance with the “Guidelines on the Utilization of Human Embryonic Stem Cells” of MEXT of Japan.

### 4.2. Immunochemical Analysis

The immunochemical experiments were conducted following the procedures described in our previous work [13]. In brief, samples were fixed with 4% paraformaldehyde (FUJIFILM Wako Pure Chemical Industries) in phosphate-buffered saline (PBS; Thermo Fisher Scientific) for 20 min, permeabilized with 0.1% Triton X-100 (FUJIFILM Wako Pure Chemical Industries) in PBS for 10 min, and blocked with 4% Block Ace (DS Pharma Biomedical, Osaka, Japan) and 0.1% Triton X-100 in PBS for 1 h at room temperature. The samples were incubated with primary antibodies diluted in Can Get Signal Solution 1 (TOYOBO, Osaka, Japan) overnight at 4 °C. The next day, the cells were washed three times with 0.1% Triton X-100 in PBS, and then were incubated for 2 h at RT with secondary antibodies (anti-mouse Alexa Fluor-488 and anti-rabbit Alexa Fluor-555; 1:1000; Thermo Fisher Scientific) diluted in Can Get Signal 2 solution (TOYOBO). The primary antibodies used were as follows: mouse anti-NGFR (1:200; Advanced Targeting Systems, San Diego, CA, USA), mouse anti-PHOX2B (1:100; Proteintech, Rosemont, IL, USA), mouse anti-DBH (1:50; Santa Cruz Biotechnology, Dallas, TX, USA), rabbit anti-PRPH (1:1000; Merck Millipore, Darmstadt, Germany), rabbit anti-PHOX2B (1:100; Abcam, Cambridge, UK), and rabbit anti-CHAT (1:1000; Abcam). To stain the nuclei, 0.2 μg/mL Hoechst 33342 (DOJINDO LABORATORIES, Kumamoto, Japan) was added to the Can Get Signal 2 solution.

### 4.3. Real-Time Quantitative PCR

The real-time quantitative PCR experiments were conducted following the procedures described in our previous work [13]. In brief, total RNA was extracted from the samples using the ReliaPrep RNA Cell Miniprep System (Promega, Madison, WI, USA) following the manufacturer’s instructions. The purity and concentration of RNA were assessed using a NanoDrop Lite spectrophotometer (Thermo Fisher Scientific). Reverse transcription of cDNA from 100 ng of total RNA was performed using the ReverTra Ace qPCR RT Kit (TOYOBO). qRT-PCR was performed on the PikoReal 96 Real-Time PCR system (Thermo Fisher Scientific) using THUNDERBIRD SYBR qPCR Mix (TOYOBO). The expression levels were normalized to the reference gene *36b4* and are presented as mean ± standard deviation (SD) values of triplicate measurements. The following primers were used: *SOX10*, forward 5′- CCTCACAGATCGCCTACACC-3′, reverse 5′- CATATAGGAGAAGGCCGAGTAGA-3′; *PRPH*, forward 5′- GCCTGGAACTAGAGCGCAAG-3′, reverse 5′- CCTCGCACGTTAGACTCTGG-3′; *PHOX2A*, forward 5′-TCGCTGAGACCCACTACCC-3′, reverse 5′-CCTGTTTGCGGAACTTGG-3′; *PHOX2B*, forward 5′-GCTGGCCCTGAAGATCGAC-3′, reverse 5′-TCAGACTTTTTGCCCGAGGAG-3′; *TH*, forward 5′-GCGCAGGAAGCTGATTGC-3′, reverse 5′-CAATCTCCTCGGCGGTGTAC-3′; *36B4*, forward 5′-AGATGCAGCAGATCCGCA-3′, reverse 5′-GTTCTTGCCCATCAGCACC-3′.

### 4.4. Viral Infection

To visualize the induced neurons from human PSCs, we employed a lentiviral vector containing green fluorescent protein (GFP) cDNA, neomycin resistance genes, and the human Synapsin-1 promoter (Synapsin-1-GFP) [41]. The cells were exposed to the Synapsin-1-GFP vector for a 24 h incubation, after which they were transferred to fresh culture medium without the vector.

### 4.5. Microelectrode Array (MEA) Recording

MEA recordings of neurons induced from human PSCs were performed using a Muse recording system (Axion Biosystems, Atlanta, GA, USA). Induced neurons were plated and cultured onto MEA substrates with embedded 64 electrodes (M64-GL1-30Pt; Axion Biosystems). Extracellular voltage signals were recorded at a sampling rate of 25 kHz and filtered with a bandpass filter (0.2–3 kHz).

### 4.6. RNA-seq

RNA-seq of induced EBs was performed according to the method previously described [13]. Briefly, total RNA was isolated from the EB samples on days 0, 7, and 13 using the ReliaPrep RNA Cell Miniprep System (Promega). The purity and concentration of RNA were checked using a NanoDrop Lite spectrophotometer (Thermo Fisher Scientific, Waltham, MA, USA). Subsequent RNA-seq operations were performed at Macrogen (Seoul, Korea). The library was constructed using over 1 μg of cDNA, and qualitatively assessed using the Agilent 2100 Bioanalyzer (Agilent Technologies, Santa Clara, CA, USA). The resultant library was sequenced with a NovaSeq 6000 (Illumina, San Diego, CA, USA) to produce 150-bp paired-end reads. The obtained reads were mapped to the human hg38 genome, trimming 10 bases from the 5′ end and 80 bases from the 3′ end using hisa2 (ver. 2.1.0). The mapped reads were converted to fragments per kilobase of exon per million mapped sequence reads (FPKM), using cufflinks (ver. 2.2.1) and gene annotation data provided by Illumina iGenome. The FPKM values were quantile normalized using an in-house R script for subsequent analysis using RStudio (ver. 2022.12.0) and R-4.4.2. Genes were included in the analysis if their FPKM values were equal to or greater than 1 in any of the samples. A DEG was defined if its FPKM fold-change was more than 2 between conditions. The MA plot for the upregulated or downregulated DEGs was created with RStudio using an in-house R script. The raw sequences in FASTQ format are available from the DNA data bank of Japan (DDBJ) (DRA008963).

### 4.7. Enrichment Analysis

For enrichment analysis of the DEGs, a list containing official gene symbols of the DEGs was submitted to the Database for Annotation, Visualization, and Integrated Discovery to conduct enrichment analysis for GO_BP, KEGG pathways [36], Reactome pathways [37], and Wikipathways [38]. Terms with adjusted *p*-values of <0.05 were considered significantly enriched with DEGs. For visualization and comparison, a negative log transformation of adjusted *p*-values was used. Bar graphs for enriched terms were created using Microsoft Excel for Microsoft 365 MSO.

A GESA was conducted using GSEA desktop software (ver. 4.3.2, Broad Institute, Cambridge, MA, USA). The DEG list calculated from as above was submitted to GSEA software, and collected gene sets and molecular signatures (c2.all.v2022.1.Hs.symbols.gmt and c5.all.v2022.1.Hs.symbols.gmt) obtained from the Molecular Signatures Database [42] were used to analyze the data. Gene sets with a nominal *p*-value of <0.05 were considered significant.

### 4.8. PPI Network Construction and Module Analysis

The network of interactions among the protein encoded by DEGs was constructed and visualized using Cytoscape 3.9.1 through the Search Tool for the Retrieval of Interacting Genes (STRING) online database [43]. We selected the PPI relation pairs with a medium confidence (interaction score cutoff was 0.4). Then, MCODE, a Cytoscape plugin, was used to identify the key gene clusters in the PPI network with the below parameters: minimum node number = 10, degree cut-off = 2, node score cut-off = 0.2, k-core = 2, and max depth = 100. The top four MCODE score gene clusters were selected and used for further analysis. Topological analysis was utilized to identify hub genes in the gene cluster networks with cytoHubba [44], another Cytoscape plugin. The “Degree” method was used to determine hub genes. Detected genes, including hub genes, in each gene cluster were visualized using “Attribute Circle Layout”. Functional gene enrichment analysis for the identified gene cluster was also conducted using the STRING enrichment plugin for Cytoscape. Terms with an adjusted *p*-value of <0.05 were considered significant. For comparison, a negative log transformation of adjusted *p*-values was used.

### 4.9. Construction of ANS-Associated Network

To explore candidate genes for autonomic induction, a PPI network for autonomic development-associated genes was constructed. Seven significant autonomic marker genes (*ASCL1*, *DDC*, *GATA2*, *GATA3*, *PHOX2A*, *PHOX2B*, and *TH*) were selected, as we previously reported [13], and were imported into NetworkAnalyst 3.0 [45], which is a comprehensive web-based platform for network-based analysis of gene expression profiling, using STRING database. The generated dataset was imported to Cytoscape for further analysis. The ANS-associated network was constructed by extracting the gene sets included in upregulated DEGs in day 13 samples compared to day 0 samples from the imported dataset. Nodes with over 2 degrees (edges) were selected and visualized.

### 4.10. TF Activity Analysis

The activity of TFs was inferred from their effects on the expression of the TF target genes. *ASCL1* and *PHOX2B* were selected for autonomic TFs, and *NEUROD1* and *POU4F1* for sensory TFs, as they are critical master regulators [25]. First, TF target genes, whose TF-binding sites are experimentally identified, were preferentially obtained through the TFLink database [46]. In addition, several TF target genes were predicted and selected using the iRegulon plugin [47] in Cytoscape. Changes in the FPKM expression patterns of TF target genes were calculated and visualized with RStudio, using an in-house R script.

### 4.11. Calcium Imaging

The induced neurons were functionally examined using calcium imaging. Samples were labelled with 10 µM Fluo-8/AM (AAT Bioquest, CA, USA) for 30 min. After labelling, the culture medium was replaced with Ringer’s solution (148 mM NaCl, 2.8 mM KCl, 2 mM CaCl_2_, 1 mM MgCl_2_, 10 mM HEPES, and 10 mM glucose; pH 7.4). The sample was transferred onto the stage of an inverted microscope (IX81; Olympus, Tokyo, Japan). The fluorescence was detected with a confocal spinning disk confocal laser scanning unit (CSC-W1; Yokogawa Electric., Tokyo, Japan) and an LED light (X-Cite 120LED; OPTO SCIENCE, Tokyo Japan). For drug application, L-glutamate (10 µM; in PBS(-); FUJIFILM Wako Pure Chemical Industries), capsaicin (10 µM; in ethanol; Sigma-Aldrich), menthol (10 µM; in ethanol; Sigma-Aldrich) and nicotine (10 µM; in ethanol; Sigma-Aldrich) were added to Ringer’s solution during fluorescence observation. A frame rate of 1.25 /s was used. The recorded fluorescence signals were analyzed using ImageJ software [48] (ver. 1.54d, National Institutes of Health, Bethesda, MD, USA; available at https://imagej.net/ij/index.html (accessed on 18 May 2023)).

### 4.12. Statistical Analysis

All data are expressed as mean ± SDs. Differences between experimental groups were analyzed with an unpaired Student’s *t*-test. Differences with *p*-values < 0.05 were considered statistically significant.

### 4.13. Data Availability

Bulk RNA sequencing data are deposited on DDBJ under the accession number DRA008963. All other data that support the findings of this study are available from the corresponding authors on request.

## Figures and Tables

**Figure 1 ijms-24-09053-f001:**
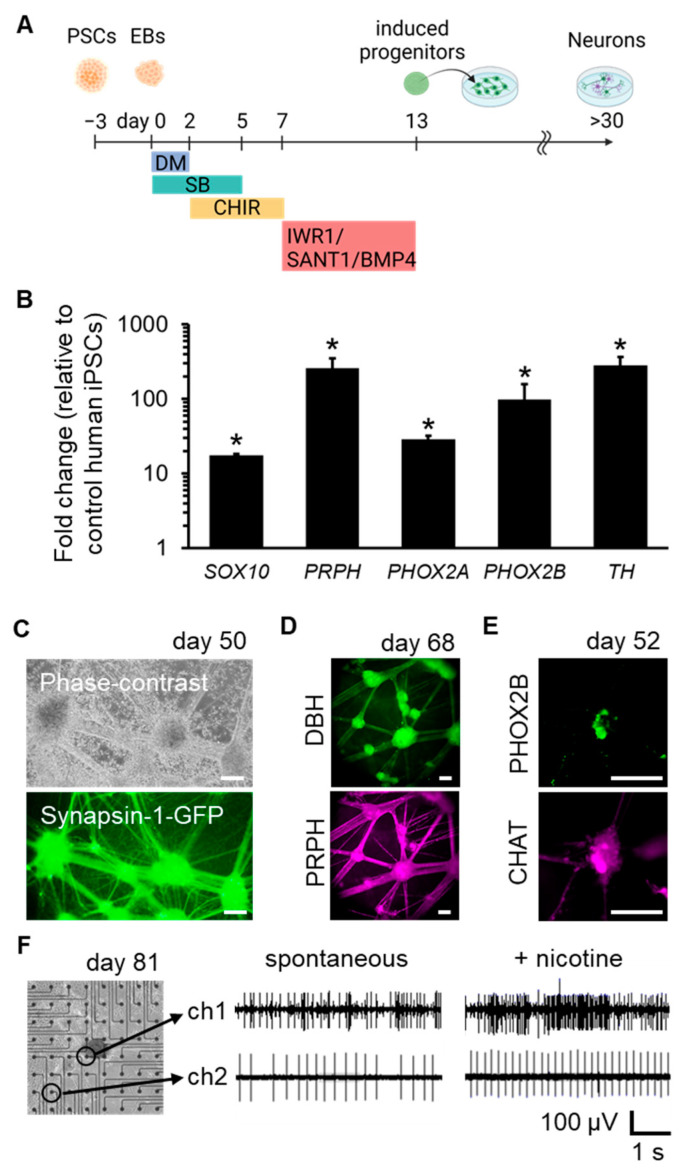
Derivation of sympathetic and parasympathetic neurons from human PSCs. (**A**) Schematic representation of autonomic induction of human PSCs. (**B**) mRNA expression levels on day 13 (n = 3; error bars represent standard deviations (SDs); two-sided Student’s unpaired t-test. * *p*  <  0.01). (**C**) Phase-contrast and fluorescent images of induced neurons on day 50, labelled with a Synapsin-1-GFP reporter to visualize neuronal networks. (**D**) Immunofluorescent staining of induced neurons for sympathetic neuron marker DBH and peripheral neuron marker PRPH on day 68. (**E**) Immunofluorescent staining of the induced neurons for PHOX2B and parasympathetic neuron marker CHAT on day 52. (**F**) Representative electrical signals of induced neurons cultured on microelectrode array (MEA) substrates on day 81. Signals from two selected electrodes before and after nicotine application are shown. Scale bar; 100 µm. EBs: embryoid bodies.

**Figure 2 ijms-24-09053-f002:**
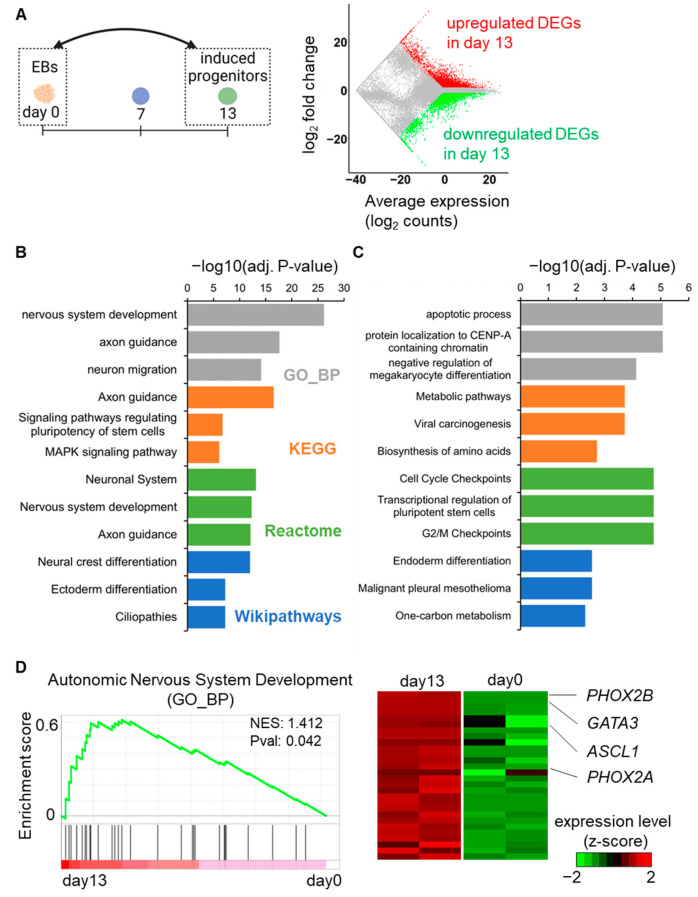
Transcriptome analysis of DEGs on day 13 compared to day 0. (**A**) Global mRNA expression patterns between day 13 and day 0. The MA plot shows upregulated (red color) and downregulated (green color) DEGs on day 13. (**B**) Bar graph representations of the top three enriched GO_BP terms (grey bars), KEGG pathways (orange bars), Reactome pathways (green bars), and Wikipathways (blue bars) for the upregulated DEGs according to adjusted *p*-values. (**C**) Bar graph representations of the top three enriched GO_BP terms (grey bars), KEGG pathways (orange bars), Reactome pathways (green bars), and Wikipathways (blue bars) for the downregulated DEGs according to adjusted *p*-values. (**D**) GSEA demonstrates significant enrichment for autonomic development-associated genes on day 13. The heatmap displays the expression changes of genes in the enrichment plot.

**Figure 3 ijms-24-09053-f003:**
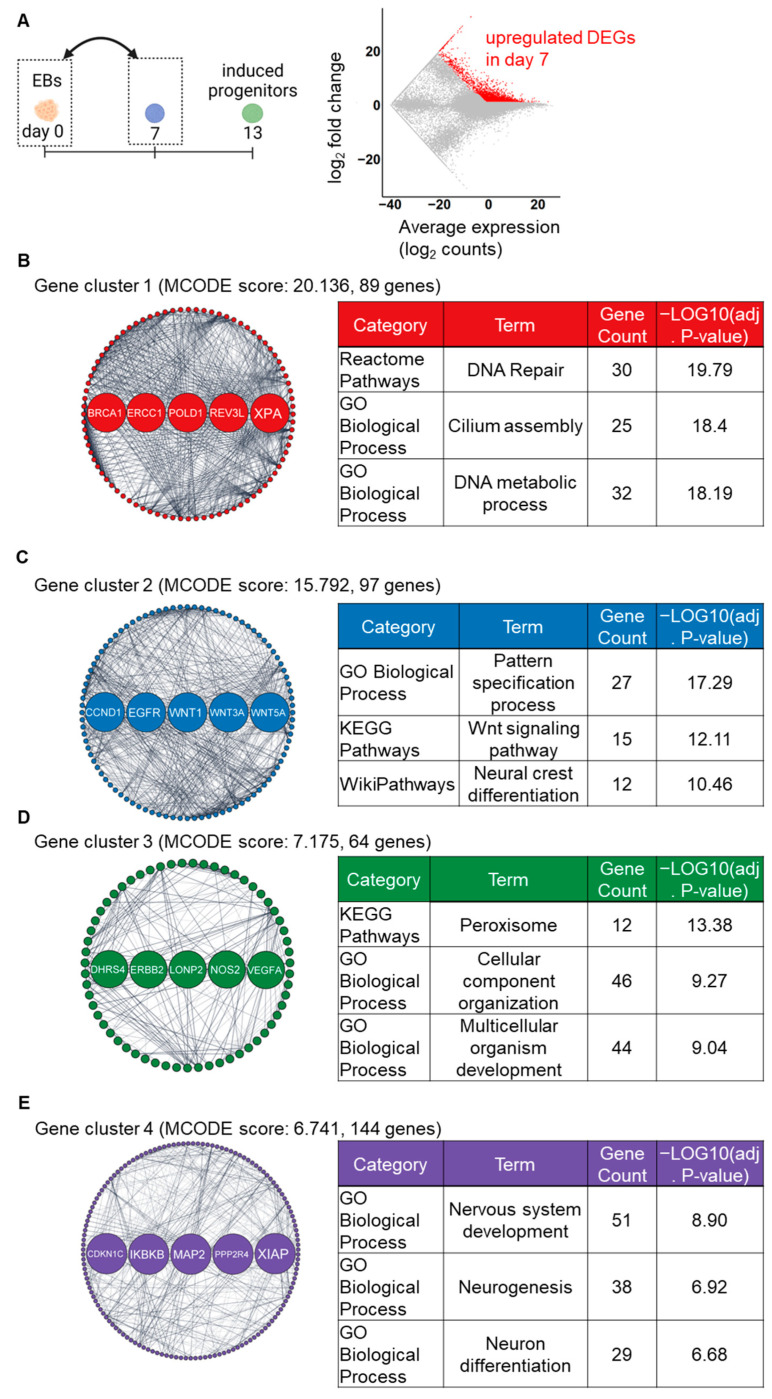
GRN-based analysis identifying the major gene clusters, hub genes, and enriched pathways on day 7 compared to day 0. (**A**) The MA plot shows upregulated (red color) DEGs on day 7. (**B**–**E**) The top four gene clusters in the PPI network of DEGs, as identified by the MCODE plugin. Highlighted nodes indicate the five hub genes in each gene cluster, as identified by the cytohubba plugin. Tables show the representative enriched pathways in each gene cluster.

**Figure 4 ijms-24-09053-f004:**
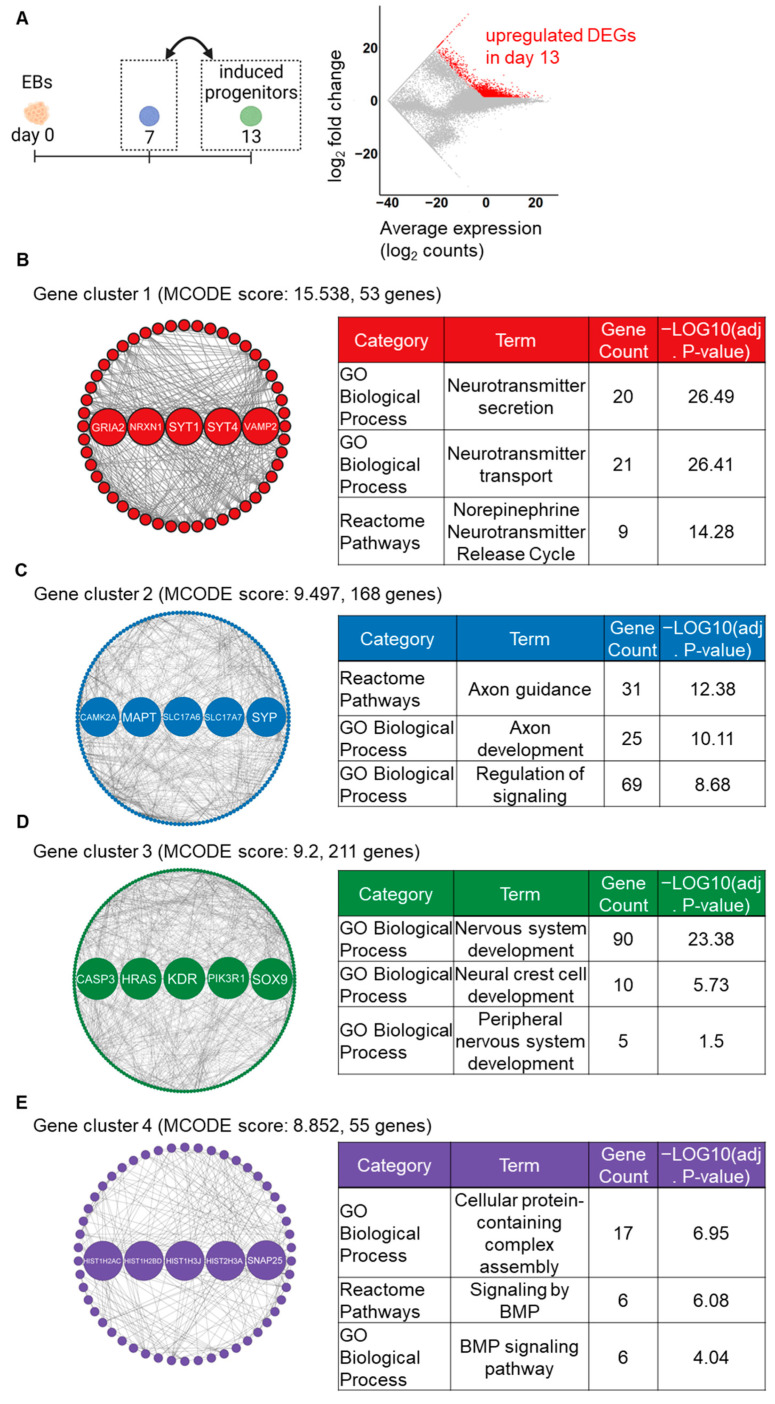
GRN-based analysis identifying major gene clusters, hub genes, and enriched pathways on day 13 compared to day 7. (**A**) The MA plot shows upregulated (red color) DEGs on day 13. (**B**–**E**) The top four gene clusters in the PPI network of DEGs, as identified by the MCODE plugin. Highlighted nodes indicate the five hub genes in each gene cluster, as identified by the cytohubba plugin. Tables show the representative enriched pathways in each gene cluster.

**Figure 5 ijms-24-09053-f005:**
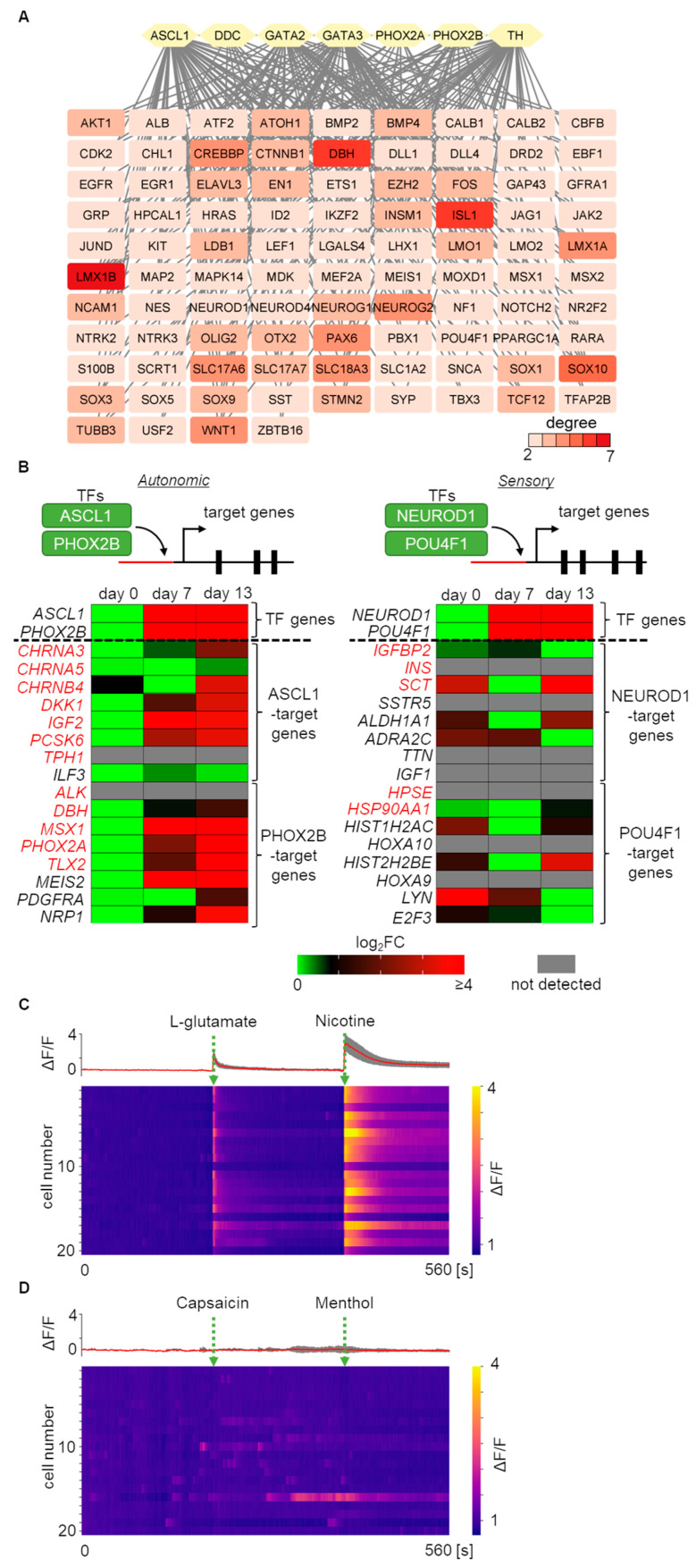
GRN and bioinformatics-based analysis identifying regulatory candidate genes and activity during autonomic induction. (**A**) Identification of the core gene network involved in autonomic induction processes. In total, 101 autonomic development-associated genes, including the 7 input autonomic marker genes, are listed. Nodes with over two degrees are visualized. (**B**) Regulatory activity of autonomic and sensory TFs during autonomic induction. Changes in gene expression of TFs and their target genes are displayed. Genes marked in red indicate experimentally identified interactions with TFs. (**C**) Calcium signal heatmap of 20 induced neurons exposed to L-glutamate and nicotine. The upper panel displays overlapped (grey) and averaged (red) traces of calcium transients at day 33. (**D**) Calcium signal heatmap of 20 induced neurons exposed to capsaicin and menthol. The upper panel displays overlapped (grey) and averaged (red) traces of calcium transients at day 33.

## Data Availability

mRNA sequencing data were deposited in DDBJ under the accession number DRA008963. All other data supporting the findings of this study are available from the corresponding authors upon request.

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
