# Peer review of "Deciphering the Molecular Mechanisms of Autonomic Nervous System Neuron Induction through Integrative Bioinformatics Analysis"

_ijms, 2023, doi:10.3390/ijms24109053_

Round 1
Reviewer 1 Report
I analysed the manuscript by Takayama et al, entitled 'Deciphering the molecular mechanisms of autonomic nervous
system neuron induction through integrative bioinformatics
analysis' and I consider that is a valuable work nicely presented. However, I have a few concerns that should be addressed before publication:
1. There are many abbreviations in the text of the manuscript and only parte of them are detailed. This issue should be revised and maybe an abbreviation list would be helpful for the reader.
2. Reading the discussion, in the end it is not clear what would be the usefulness of such induced autonomic neurons in research for deciphering human pathology and for identifying treatment targets
3. A comparison with human autonomic neurons transcriptom is not proved or mentioned.
Author Response
We greatly appreciate the reviewer’s comments which have helped us to improve the quality of our paper.
Comment1
There are many abbreviations in the text of the manuscript and only parte of them are detailed. This issue should be revised and maybe an abbreviation list would be helpful for the reader.
Our response:
We greatly appreciate the reviewer’s suggestion and apologize for the omission.
We have added descriptions for some abbreviations and an abbreviation list before Reference section of the revised manuscript.
Comment2
Reading the discussion, in the end it is not clear what would be the usefulness of such induced autonomic neurons in research for deciphering human pathology and for identifying treatment targets.
Our response:
We greatly appreciate the reviewer’s suggestion and apologize for the omission.
Induced autonomic neurons have myriad potential applications in biomedical research. The most immediately accessible application lies in their ability to facilitate disease modeling, contributing significantly to our understanding of disease pathophysiology. This, in turn, propels advancements in drug discovery and validation. Furthermore, these neurons can be leveraged in the fields of regenerative medicine and gene therapy research, acting as valuable therapeutic tools or resources. Collectively, these applications have substantial implications for the evolution of personalized medicine, shaping a future where treatments are tailored to individual patient profiles.
For use as research tools in modeling, the neurons would allow construction of co-culture systems with various target organs. This could potentially reconstruct homeostatic functions and inter-organ interactions in vitro. We think that this approach could further contribute to unraveling the mechanisms of diseases involving the ANS and could be applied to drug screening.
We have added these descriptions to the discussion section of the revised manuscript.
Comment3
A comparison with human autonomic neurons transcriptome is not proved or mentioned.
Our response:
We greatly appreciate the reviewer’s suggestion.
The major problem in human ANS induction research lies in the difficulty of obtaining human autonomic neurons. This is thought to be due to the autonomic neurons being dispersed throughout the body with limited cell numbers, as well as ethical considerations. However, as the reviewer rightly point out, it is crucial to compare and analyze with the transcriptome of actual human autonomic neurons. We believe it is important to consider whether it is possible to obtain and analyze human autonomic neurons through collaborations with medical departments around the world.
We have added this description in the Discussion section of the revised manuscript.
Reviewer 2 Report
In this experimental study of iPSC generated autonomic nervous system (ANS)
progenitors the authors performed high-throughput RNA-sequencing to explore
the transcription factors (TF) involved in this process. Their gene regulatory
network (GRN) analysis provide insights on the gene clusters involved seven
days after the propagation of autonomic nervous system neurons from iPSCs.
The authors evaluated their protocol with biomarkers and functionally by
calcium signalling.
Minor concerns
1. It would be preferable to add some more details on the TFs and genes
involved in this process.
Author Response
We greatly appreciate the reviewer’s comments which have helped us to improve the quality of our paper.
Comment1
It would be preferable to add some more details on the TFs and genes involved in this process.
Our response:
We greatly appreciate the reviewer’s suggestion.
For definition of induced ANS progenitors, analysis of the expression of TFs ASCL1 and PHOX2B is important. In particular, PHOX2B is a crucial marker for ANS progenitors, known to regulate the expression of TH and DBH, which are sympathetic neuron markers, and CHAT, a parasympathetic neuron marker.
We have added these descriptions in the Introduction section of the revised manuscript.
Reviewer 3 Report
1) Results: how statistical significance s reported (based on Materials & Methods as p-value but based on Tableas a -LOG)?
2) Discussion should be improved
3) More data about "Bionifromatics tools/software for analysis and visualization will be beneficial
Minor editing of English language required
Author Response
We greatly appreciate the reviewer’s comments which have helped us to improve the quality of our paper.
Comment1
Results: how statistical significance s reported (based on Materials & Methods as p-value but based on Tableas a -LOG)?
Our response:
We thank the reviewer for this comment and apologize for the omission.
Statistical analysis was first conducted by calculating the P-values to detect significant differences. Subsequently, for clarity in comparisons such as the bar graphs in Fig.2B or the tables in Figs. 3 and 4, we performed a negative log transformation of P-values and used it in the figures and tables.
We have added this description in the Materials and Methods section of the revised manuscript.
Comment2
Discussion should be improved
Our response:
We thank the reviewer for this comment.
Considering the opinions of other reviewers, we have added further details about the utility of the induced neurons and the limitations of our study in the discussion section of the revised manuscript.
Comment3
More data about "Bionifromatics tools/software for analysis and visualization will be beneficial
Our response:
We greatly appreciate the reviewer’s suggestion.
We have added some details and descriptions in the Materials and Methods sections of the revised manuscript.